# Socio-economic factors associated with cancer stigma among apparently healthy women in two selected municipalities Nepal

**Bandana Paneru**[1,2]*, **Aerona Karmacharya**[1], **Soniya Makaju**[1], **Diksha Kafle**[3], **Lisasha Poudel**[2], **Sushmita Mali**[1], **Priyanka Timsina**[1], **Namuna Shrestha**[1], **Dinesh Timalsena**[1], **Kalpana Chaudhary**[1], **Niroj Bhandari**[2,4], **Prasanna Rai**[1], **Sunila Shakya**[5], **Donna Spiegelman**[6,7], **Sangini S. Sheth**[8], **Anne Stangl**[9], **McKenna C. Eastment**[10], **Archana Shrestha**[1,2,6]

1 Department of Public Health and Community Programs, Kathmandu University School of Medical Sciences, Dhulikhel, Nepal, 2 Institute for Implementation Science and Health, Kathmandu, Nepal, 3 Canadian Red Cross, Country Office Nepal, Kathmandu, Nepal, 4 Authentic Leadership Institute Nepal, Kathmandu, Nepal, 5 Department of Obstetrics and Gynecology, Dhulikhel Hospital/Kathmandu University School of Medical Sciences, Dhulikhel, Nepal, 6 Center of Methods for Implementation and Prevention Science, Yale School of Public Health, New Haven, CT, United States of America, 7 Department of Biostatistics, Yale School of Public Health, New Haven, CT, United States of America, 8 Department of Obstetrics, Gynecology, and Reproductive Sciences, Yale School of Medicine, New Haven, CT, United States of America, 9 International Center for Research on Women, Washington, DC, United States of America, 10 Division of Allergy and Infectious Disease, Veterans Affairs Puget Sound Healthcare System and University of Washington, Seattle, WA, United States of America

* bandana.paneru@gmail.com

## Abstract

### Introduction

Cancer is the primary cause of death globally, and despite the significant advancements in treatment and survival rates, it is still stigmatized in many parts of the world. However, there is limited public health research on cancer stigma among the general female population in Nepal. Therefore, this study aims to determine the prevalence of cancer stigma and its associated factors in this group.

### Methods

We conducted a cross-sectional study among 426 healthy women aged 30 to 60 years who were residents of Dhulikhel and Banepa in central Nepal. We measured cancer stigma using the Cancer Stigma Scale (CASS). CASS measures cancer stigma in six domains (awkwardness, avoidance, severity, personal responsibility, policy opposition, financial discrimination) on a 6-point Likert scale (strongly disagree to agree strongly) with higher mean stigma scores correlating with higher levels of stigma. We utilized Generalized Estimating Equations (GEE) with multivariable linear regression to identify the socio-demographic factors associated with the CASS score.

**Data Availability Statement:** Data are available within the manuscript as a supporting Information files

**Funding:** This study was funded by the National Cancer Institute Support Grant (NCI-P30CA016359) and awarded to Spiegelman D. and Shrestha A.

**Competing interests:** The authors have declared that no competing interests exist

## Results

Overall, the level of cancer stigma was low, with a mean stigma score of 2.6 (0.6), but it was still present among participants. Stigma related to personal responsibility had the highest levels, with a mean score of 3.9 (1.3), followed by severity with a mean score of 3.2 (1.3), and financial discrimination with a mean score of 2.9 (1.6). There was a significant association between the mean CASS score and older age (mean difference in stigma score: 0.11 points; 95% CI: 0.02–0.20) as well as lower education (difference: -0.02 points; 95% CI: -0.03 to -0.003), after adjusting for age, ethnicity, education, marital status, religion, occupation, and parity.

## Conclusion

While overall cancer stigma was low, some domains of stigma were higher among women in a suburban area in central Nepal; thus, indicating that cancer stigma persists in this region despite its low overall prevalence.

## Introduction

Cancer is the leading cause of death worldwide, with age-standardized mortality of 100 per 100,000 population in 2020 [1]. The majority of cancer deaths occur in low- and middle-income countries [1]. In Nepal, the age-standardized cancer incidence rate is 80 per 100,000 with a mortality rate of 54 per 100,000 [1]. Despite recent improvements in treatment and survival, cancer is still a stigmatized disease, [2–4] and one of the most feared illnesses [5].

Health-related stigma subjects a person or group to exclusion, rejection, blame, or devaluation due to the anticipation or experience of negative social judgment regarding their health condition, making it a social phenomenon or personal experience [6]. Public stigma may appear in the form of stereotypes, for instance, viewing individuals who have survived cancer as either incapable or contagious. These stereotypes result in behaviors like avoiding interactions with cancer survivors due to a fear of contracting the disease. Discriminatory actions stemming from these biases can lead to withholding job opportunities or rejecting social interactions [7–9]. This public stigma significantly hinders individuals' willingness to seek health care [6, 10]. Prior research has predominantly examined stigma related to illnesses such as leprosy [11, 12] epilepsy, HIV/AIDS [13, 14] and mental illness [15–17]. Although people often stigmatized cancer, there is a limited exploration of public stigma in healthy general female populations [10]. Examining public stigma related to cancer is crucial for several reasons. First, stigma could dissuade individuals from participating in cancer prevention and screening efforts, resulting in delayed cancer diagnoses and, ultimately, higher mortality rates [10, 18–20]. Second, public health initiatives aimed at educating people about the behavioral factors linked to cancer, including smoking, obesity, and infection with the human papillomavirus, could create stigma by suggesting that cancer is avoidable and depends on individual behavior and choices [21]. Third, stigma can contribute to health disparities, [22–24] particularly among marginalized groups who may already face barriers to accessing healthcare.

Few studies have actively investigated cancer-related stigma and associated socio-demographic factors. Qualitative research delving into experienced and internalized stigma among cancer patients reveals that young, single individuals encounter distinct stigma experiences influenced by their age, gender, marital status, socio-economic position, and family living

arrangements [25, 26]. A quantitative exploration of stigma in England revealed a higher prevalence of cancer stigma among men and individuals from ethnic minority backgrounds but no associations with age or social status. Notably, these studies utilized only 18 of the 25 CASS available items, potentially resulting in low internal reliability. Moreover, studies have yet to be conducted in low-income countries, notably Nepal. Our team previously conducted two studies on cancer stigma. The first assessed the validity of the Cancer Stigma Scale (CASS) in the Nepalese context, and the second examined the relationship between cancer stigma and cervical cancer screening uptake, finding that women with higher stigma were less likely to participate in screening. To better understand this, it is important to examine the level and distribution of stigma, identify the domains where stigma is most prevalent, and analyze which socio-economic groups are more affected. Therefore, in this study, we aim to determine item-specific, domain-specific, and overall cancer stigma scores, as well as assess the socio-economic factors associated with cancer stigma. Given the limited research on socio-economic influences on cancer stigma, we selected these factors based on a review of prior literature [10, 20].

Our team previously conducted two studies on cancer stigma. The first assessed the validity of the Cancer Stigma Scale (CASS) in the Nepalese context, [27] and the second examined the relationship between cancer stigma and cervical cancer screening uptake, finding that women with higher stigma were less likely to participate in screening [28]. To better understand this, it is important to examine the level and distribution of stigma, identify the domains where stigma is most prevalent, and analyze which socio-economic groups are more affected. Therefore, in this study, we aim to determine item-specific, domain-specific, and overall cancer stigma scores, as well as assess the socio-economic factors associated with cancer stigma. Given the limited research on socio-economic influences on cancer stigma, we selected these factors based on a review of prior literature [10, 20] The novelty of this study lies in its focus on quantitatively measuring cancer stigma within the non-cancer population, a topic that has been sparsely explored in previous research. Additionally, prior studies have not provided detailed assessments of stigma associated with individual items and domains. This information will be valuable in designing targeted stigma reduction interventions by identifying specific areas that need attention. By identifying these factors, we can develop interventions and policies that can help reduce stigma and its adverse effects on individuals and society and ultimately improve the population's health, where cancer screening and early treatment are lifesaving.

## Methods

### Study design and setting

We conducted a cross-sectional study in two municipalities of the Kavrepalanchow district of Nepal, Dhulikhel and Banepa, approximately 30 kilometers east of Kathmandu. Dhulikhel is a semi-urban location with a population of 32,162, while Banepa has a population of 55,628. The dominant ethnic group in both municipalities is the Newar, and 70% of the population is literate. The literacy rate among females is around 75% in both municipalities [29].

### Study participants

Our study population included women aged 30 to 60 years who were residents of Dhulikhel or Banepa. The exclusion criteria were: a) having a hearing impairment, b) having severe mental health conditions so as being not able to provide informed consent, and c) not being a resident of Dhulikhel or Banepa (i.e., visitors to the area).

## Recruitment

We enrolled the initial 426 women out of 1800 who underwent cervical cancer screening organized by Dhulikhel Hospital from May 15 to September 15, 2021. We estimated the sample size based on an expected proportion of cancer stigma among women of 51%,(10) at a 5% significance level, and a margin of error of 5% [30].

Our Research Assistants (RAs) contacted Female Community Health Volunteers (FCHV) and oriented them to the study objective and expectations. These FCHVs have worked in Nepal since the 1980s and play a pivotal role in the Nepali community health workforce, specializing in health education, counseling, outreach, and resource distribution [31]. The FCHVs disseminated information about the study to women in their network and shared the contact details of the interested participants with the research assistants. Subsequently, the research assistants contacted the women by phone, outlining the study's objectives and explaining their potential role. Women expressing interest were then formally enrolled in the study after verbal informed consent was obtained, ensuring the anonymity and confidentiality of their information. We conducted the study amid the COVID-19 pandemic. Therefore, interviews were conducted over the phone for infection prevention. Kathmandu University Institutional Review Committee ethical board approved the study (KUIRC no: 35/2021).

## Data collection

Trained research assistants conducted telephonic interviews using a structured questionnaire directly entered electronically (Kobo toolbox) [32]. The questionnaire covered socio-demographic factors and cancer stigma.

## Measures

**Cancer stigma.**    We used Nepal's validated CASS tool [27]. The CASS demonstrated satisfactory internal reliability, with a Cronbach's alpha of 0.88 for the overall scale and ranging from 0.70 to 0.89 for its six components. [27] The CASS consists of 25 items that assess six domains: (a) awkwardness-5 items, which measures how comfortable people feel around someone with cancer; (b) severity-5 items, which evaluates the expected severity of cancer consequences and the likelihood of recovery; (c) avoidance-5 items, which assesses the extent to which people avoid cancer patients and maintain physical distance from them; (d) personal responsibility-4 items, which determines how much a person's actions contribute to their cancer; (e) policy opposition-3 items, which gauges the perceived responsibility of the government and the public in the care and treatment of cancer patients; and (f) financial discrimination-3 items, which measures the anticipated deprivation of benefits to cancer patients from banking and insurance services.

Participants responded using a 6-point Likert scale ranging from 'disagree strongly' to 'agree strongly.' We reversed the scores for five specific items, ensuring higher scores reflected more stigma levels (refer to Table 2 for details on the reverse-scored items). The mean score for each domain was then calculated [10, 17].

**Socio-demographic variables.**    Socio-demographic variables included age (in years), ethnicity (Brahmin/ Chettri/Thakuri/Sanyasi, Newar, Magar/Tamang/Rai/Limbu, Sherpa/Bhote, Kami/Damai/Sarki/Gaaine/Baadi, Other), education (number of years of formal education completed), religion (Hindu, Buddhist, Christian), occupation (homemaker, farmer, business, unemployed, others) and parity (number of children). We adopted the questions from previously conducted national surveys in Nepal. Responses were self-reported [33, 34].

**Data analysis.** We calculated summary statistics such as frequency and percentage for categorical variables and mean with Standard Deviation (SD) for continuous variables. We estimated cancer stigma scores for six sub-domains and calculated mean scores for each subscale.

We checked the assumptions of linearity, independence, normality, and equal variance for linear regression in our analysis. We used a scatterplot of residuals versus predicted values to assess the linearity and equal variance assumptions. To check for normality, we examined a histogram and a normal probability plot of the residuals. We used the GEE model to address clustering (dependence) by ward. Additionally, to detect multicollinearity, we used the variance inflation factors (VIF) method. The mean VIF was around 2.10, indicating that multicollinearity was not problematic.

We utilized GEE with multivariable linear regression, exchangeable working correlation, and robust variance to account for the clustering effect of community people on stigma scores, using the ward level as the clustering level. Variables were selected for multivariable regression analysis based on previous literatures [10, 20] and prior knowledge. We reported crude and adjusted differences in stigma scores and their 95% confidence intervals and p-values. For age variables, we presented coefficients in 10-year increments to enhance interpretability, as one-year changes yield small, less meaningful coefficients, while 10-year groupings capture more substantial age-related trends.

We conducted all analyses using STATA-13. For age variables, we have have presented coefficients in 10 years difference

## Results

Table 1 presents the socio-demographic characteristics of the participants. The mean age was 42(8) years. Most participants (43%) were Brahmin/Chhetri and Hindus (87.8%). About a third of the participants (31%) had no formal education, and the majority (39.5%) were farmers. The mean number of children was 2.3 (1.0).

Table 2 presents the CASS mean score for each stigma domain. The overall mean total CASS score was 2.6 (0.6). Within the six domains, the highest stigma level pertained to personal responsibility, reflecting the belief that patients are accountable for acquiring cancer with a stigma score of 3.9(1.3). Additionally, high stigma levels were observed in the severity domain, indicating that patients may struggle to return to everyday life, adversely affecting their overall life and relationships with a mean stigma score of 3.2 (1.3). Financial discrimination, manifested by the denial of loans and mortgage applications for cancer patients, also exhibited a notable stigma level with a mean stigma score of 2.9 (1.6). Policy opposition showed a low mean stigma score of 1.3 (0.6), suggesting strong government and community support for cancer patient care.

Table 3 demonstrates the mean CASS score in each domain for selected variables. With an increase in age, there is an increase in the overall stigma score and in the severity and avoidance domains. In the awkwardness domain, stigma scores are lower among women under 40 years old compared to other groups. The overall cancer stigma score decreases with increasing education levels. Stigma scores among women with non-formal education are higher in all domains (awkwardness, severity, avoidance, and personal responsibility) except for financial discrimination. The overall mean stigma score is highest among the Sherpa and Bhote ethnic groups, particularly in the severity domain. In the personal responsibility domain, the Damai/Sarki/Gaaine/Baadi ethnic group has the highest stigma score. In the financial discrimination domain, the Newar ethnic group has the highest stigma score. Farmers have the highest stigma scores compared to women in other occupations in the overall score and in the severity and personal responsibility domains. Women engaged in business have the highest stigma score in

**Table 1. Socio-demographic characteristics of the study participants (n = 426).**

| Characteristics | Frequency (%) |
|---|---|
| Age (in years), Mean(SD) | 42.3 (8.1) |
| Ethnicity | |
| Brahmin/Chettri/Thakuri/Sanyasi | 184 (43.2) |
| Newar | 175 (41.1) |
| Magar/Tamang/Rai/Limbu | 15 (3.5) |
| Sherpa/Bhote | 27 (6.3) |
| Kami/Damai/Sarki/Gaaine/Baadi | 25 (5.9) |
| Religion | |
| Hindu | 374 (87.8) |
| Buddhist | 28 (6.6) |
| Christian | 24 (5.6) |
| Education | |
| No formal education | 132 (31.0) |
| Primary | 49 (11.5) |
| Secondary | 150 (35.2) |
| Above secondary | 95 (22.3) |
| Occupation | |
| Farmer | 168 (39.5) |
| Home-maker | 106 (24.9) |
| Business | 63 (14.8) |
| Unemployed | 7 (1.6) |
| *Other | 82 (19.2) |
| Parity (number), Mean (SD) | 2.3 (1.1) |

*Other: cleaner, helper, tailor, labor, parlor,and teacher

SD: Standard Deviation

Education(years of formal education completed) 0 = No formal education; 1/5 = primary education; 6/10 = secondary education; and 11/max = above secondary

the financial discrimination domain compared to other occupations. The overall stigma score is highest among women who have more than two children. These women also have the highest stigma scores in the awkwardness, severity, avoidance, and financial discrimination domains. In the personal responsibility domain, women with no children have the highest stigma score. The overall stigma score is highest among women who follow Buddhism. These women have the highest stigma scores in the awkwardness, severity, and financial discrimination domains.

In the univariable regression model, the mean cancer stigma score showed associations with age, with older women having a higher mean stigma score, education, with more highly educated women having a lower mean stigma score, and occupation, with farmers having higher mean stigma scores compared to business women. However, after adjusting for socio-demographic variables, the mean cancer stigma score was only associated with age and education. There is a positive association between age and cancer stigma score. When comparing two groups with a ten-year age difference, the older group had a higher cancer stigma score of 0.11 units compared to the younger group, after accounting for differences in education, ethnicity, occupation, parity, and religion (95% CI: 0.02–0.20; p-value = 0.01).There was a significant negative association between education and the mean cancer stigma score (p-value<0.001). The mean cancer stigma score was 0.02 units lower with one year more formal

**Table 2. CASS mean score in each domain among participants (n = 426).**

|  | Mean(sd) |
|---|---|
| **Awkwardness** | **2.4 (1.2)** |
| I would feel at ease around someone with cancer (R) | 2.4 (1.7) |
| I would feel comfortable around someone with cancer (R) | 2.4 (1.7) |
| I would find it difficult being around someone with cancer | 2.3 (1.7) |
| I would find it hard to talk to someone with cancer | 2.2 (1.6) |
| I would feel embarrassed discussing cancer with someone who had it | 2.7 (1.8) |
| **Severity** | **3.2 (1.3)** |
| Once you've had cancer, you're never 'normal' again | 3.2 (1.7) |
| Having cancer usually ruins a person's career | 3.4 (1.7) |
| Getting cancer means having to mentally prepare oneself for death | 3.4 (1.7) |
| Cancer usually ruins close personal relationships | 3.1 (1.7) |
| Cancer devastates the lives of those it touches | 2.8 (1.7) |
| **Avoidance** | **1.7 (0.9)** |
| If a colleague had cancer, I would try to avoid them | 1.7 (1.2) |
| I would distance myself physically from someone with cancer | 1.8 (1.4) |
| I would feel irritated by someone with cancer | 1.3 (0.8) |
| I would feel angered by someone with cancer | 1.2 (0.7) |
| I would try to avoid a person with cancer | 2.1 (1.7) |
| **Policy Opposition** | **1.3 (0.6)** |
| More government funding should be spent on the care and treatment of those with cancer (R) | 1.3 (0.8) |
| The needs of people with cancer should be given top priority (R) | 1.2 (0.5) |
| We have a responsibility to provide the best possible care for people with cancer (R) | 1.3 (0.6) |
| **Personal Responsibility** | **3.9 (1.3)** |
| A person with cancer is liable for their condition | 4.2 (1.6) |
| A person with cancer is accountable for their condition | 4.4 (1.5) |
| If a person has cancer, it's probably their fault | 3.6 (1.6) |
| A person with cancer is to blame for their condition | 3.2 (1.6) |
| **Financial discrimination** | **2.9 (1.6)** |
| It is acceptable for banks to refuse to make loans to people with cancer | 2.4 (1.8) |
| Banks should be allowed to refuse mortgage applications for cancer-related reasons | 2.6 (1.8) |
| It is acceptable for insurance companies to reconsider a policy if someone has cancer | 3.7 (1.9) |
| **Overall stigma** | **2.6 (0.6)** |

*Stigma score ranges from 1–6. Higher scores indicate a higher sigma level. (R) indicated reversed in scoring.

education among women (95% CI:-0.03, -0.01). Cancer stigma score was not significantly associated with other socio-demographic variables such as ethnicity(p-value = 0.31), occupation (p-value = 0.49), parity(p-value = 0.98), and religion (p-value = 0.22). (Table 4)

## Discussion

Our study found a low level of general cancer stigma among apparently healthy women in semi-urban Nepal. However, we observed high levels of cancer stigma in the domains of personal responsibility, severity, and financial discrimination. The cancer stigma score was lowest in the policy opposition domain. Cancer stigma scores were high among older individuals and those with lower levels of formal education.

In previous studies exploring cancer stigma, consistently low levels of stigma have been reported, though these levels vary across different domains. Similar to our findings, a recent

**Table 3. CASS mean score in each domain for selected variable (n = 426).**

| Characteristics | Overall score | Awkwardness | Severity | Avoidance | Policy Opposition | Personal Responsibility | Financial discrimination |
|---|---|---|---|---|---|---|---|
| Age (years) | | | | | | | |
| less than 40 | 2.4(0.6) | 2.3(1.3) | 2.9(1.3) | 1.5(0.7) | 1.5(0.7) | 3.8(1.3) | 2.7(1.5) |
| 40 to 49 | 2.6(0.6) | 2.5(1.3) | 3.3(1.2) | 1.7(0.9) | 1.3(0.6) | 3.9(1.3) | 3.3(1.7) |
| 50 and above | 2.7(0.5) | 2.5(1.2) | 3.5(1.2) | 1.9(1.1) | 1.3(0.5) | 4.0(1.1) | 2.9(1.5) |
| Education | | | | | | | |
| No formal education | 2.8(0.7) | 2.6(1.3) | 3.5(1.3) | 1.9(1.1) | 1.4(0.6) | 4.2(1.2) | 2.9(1.6) |
| Primary | 2.6(0.6) | 2.6(1.2) | 3.5(1.1) | 1.7(1.0) | 1.3(0.4) | 3.9(1.3) | 2.7(1.5) |
| Secondary | 2.5(0.6) | 2.3(1.3) | 3.0(1.1) | 1.5(0.7) | 1.2(0.5) | 3.7(1.3) | 3.0(1.5) |
| Above secondary | 2.4(0.5) | 2.3(1.1) | 2.9(1.2) | 1.5(0.7) | 1.3(0.6) | 3.7(1.2) | 3.0(1.7) |
| Ethnicity | | | | | | | |
| Brahmin/Chhetri | 2.6(0.6) | 2.4(1.2) | 3.2(1.3) | 1.7 (1.0) | 1.4(0.6) | 4.0(1.1) | 2.7(1.5) |
| Damai/Sarki/Gaaine/Baadi | 2.5(0.5) | 2.1(1.1) | 3.2(1.)2 | 1.6(0.8) | 1.3(0.4) | 4.7(1.2) | 2.1(1.4) |
| Magar/Tamang/Rai/Limbu | 2.2(0.6) | 2.2(1.0) | 2.5(1.4) | 1.6(0.7) | 1.2(0.3) | 4.2(1.1) | 1.7(1.1) |
| Newar | 2.6(0.6) | 2.5(1.3) | 3.2(1.2) | 1.6(0.8) | 1.3(0.5) | 3.5(1.4) | 3.4(1.6) |
| Sherpa/Bhote | 2.8(0.7) | 2.5(1.5) | 3.6 (1.4) | 1.7(0.9) | 1.3(0.5) | 4.5(1.0) | 2.9(1.8) |
| Occupation | | | | | | | |
| Business | 2.6(0.6) | 2.4(1.4) | 3.1(1.2) | 1.5(0.7) | 1.3(0.5) | 3.8(1.4) | 3.2(1.9) |
| Farmer | 2.7(0.6) | 2.4(1.3) | 3.3(1.3) | 1.7(0.9) | 1.3(0.6) | 4.1(1.2) | 3.0(1.7) |
| Home-maker | 2.5(0.6) | 2.3(1.3) | 3.0(1.3) | 1.7(0.9) | 1.3(0.5) | 3.8(1.3) | 2.8(1.4) |
| Others | 2.6(0.7) | 2.5(1.3) | 3.0(1.2) | 1.6(1.0) | 1.4(0.7) | 3.7(1.3) | 2.8(1.4) |
| Unemployed | 2.6(0.4) | 2.7(1.0) | 2.9(1.3) | 1.4(0.7) | 1.7(0.8) | 3.8(1.5) | 3.1(1.2) |
| Parity | | | | | | | |
| Zero | 2.1(0.7) | 1.9(0.7) | 2.1(0.7) | 1.5(0.7) | 1.0(0.3) | 4.1(1.4) | 2.2(1.9) |
| One to two | 2.6(0.6) | 2.4(1.3) | 3.1(1.3) | 1.6(0.8) | 1.3(0.6) | 3.9(1.3) | 2.9(1.6) |
| More than two | 2.7(0.6) | 2.5(1.3) | 3.4(1.3) | 1.8(0.7) | 1.3(0.5) | 3.9(1.)2 | 3.0(1.6) |
| Religion | | | | | | | |
| Buddhist | 2.8(0.7) | 2.6(1.4) | 3.7(1.2) | 1.7(0.8) | 1.3(0.5) | 4.4(1.1) | 3.2(1.9) |
| Christian | 2.6(0.7) | 2.1(1.2) | 3.1(1.2) | 1.6(0.9) | 1.5(0.9) | 4.8(0.9) | 2.4(1.7) |
| Hindu | 2.6(0.6) | 2.4(1.3) | 3.2(1.3) | 1.7(0.9) | 1.3(0.6) | 3.8(1.3) | 3.0(1.6) |

community-based study conducted in India [35] and a hospital-based study in Nepal [36] reported high stigma in the domains of severity, awkwardness, and personal responsibility, with the lowest levels in policy opposition. Additionally, a cross-sectional study conducted in the UK showed higher cancer stigma in the domains of personal responsibility and severity, similar to our study [10]. In contrast to women from the UK, our study population exhibited a higher total mean cancer stigma score across five sub-domains: severity, awkwardness, financial discrimination, personal responsibility, and avoidance. However, our study participants demonstrated a lower mean stigma score in the domain of policy opposition compared to English women, indicating less support for government funding toward cancer care and treatment [10, 17].

The low mean cancer stigma score observed in the policy opposition sub-domain suggests a distinct perspective in our research. Unlike the findings from the English study, [10] Nepali participants did not anticipate receiving substantial support from the government or community for cancer diagnosis and treatment. This difference can be attributed to Nepal and England's different health financing mechanisms and community structures. In Nepal, out-of-pocket healthcare expenditure is notably high (55%), [37] compared to England (12.5%) [38]. Our study participants may have experienced a greater need for government support for cancer treatment. Even though we didn't find a specific literature showing that differences in

**Table 4. Factors associated with cancer stigma score among participants (n = 426).**

| Characteristics | Univariable | | Multivariable | |
|---|---|---|---|---|
| | Difference in CASS score (95% CI) | p-value | Difference in CASS score (95% CI) | p-value |
| *Age (years), Mean (SD) | 0.20 (0.10, 0.30) | 0.002 | 0.11 (0.02, 0.20) | 0.01 |
| Education (years of formal education), Mean (SD) | 0.04 (-0.05, -0.02) | <0.001 | -0.02 (-0.03, -0.01) | <0.001 |
| Ethnicity | | | | |
| Brahmin/Chhetri | Ref | 0.67 | | 0.31 |
| Damai/Sarki/Gaaine/Baadi | 0.03 (-0.39, 0.40) | | -0.07 (-0.34, 0.24) | |
| Magar/Tamang/Rai/Limbu | 0.19 (-0.70, 0.37) | | -0.23 (- 0.49, 0.04) | |
| Newar | -0.08 (-0.29, 0.12) | | 0.04 (-0.03, 0.12) | |
| Sherpa/Bhote | 0.09 (-0.31, 0.50) | | 0.07 (-0.28, 0.43) | |
| Occupation | | | | |
| Business | Ref | 0.53 | | 0.49 |
| Farmer | 0.37(0.08, 0.67) | | 0.01 (-0.09, 0.10) | |
| Home-maker | 0.02 (-0.29, 0.33) | | -0.06 (-0.23, 0.12) | |
| Others | 0.15 (-0.18, 0.48) | | 0.04 (-0.14, 0.23) | |
| Unemployed | 0.59 (-0.20, 1.38) | | 0.07 (-0.27, 0.41) | |
| Parity, Mean (SD) | 0.09 (-0.01, 0.18) | | 0.10 (-0.24, 0.46) | 0.98 |
| Religion | | | | |
| Buddhist | Ref | 0.04 | | 0.22 |
| Christian | -0.02 (-0.57, 0.63) | | -0.13 (-0.63, 0.36) | |
| Hindu | -0.19 (-0.59, 0.19) | | -0.16 (-0.40, 0.07) | |

CI: Confidence interval

SD: Standard Deviation

Adjusting variables- age, education, ethnicity, occupation, parity and religion

*Age coefficients presented in 10 years difference

health financing affect stigma scores. It is plausible that high out-of-pocket expenditure leads participants to believe that more government funding should be spent on the care and treatment of those with cancer, resulting in lower stigma scores in policy opposition. Moreover, Nepal's more cohesive community structure might have inclined our participants to prioritize the shared responsibility of caring for cancer patients within their community. Several studies show that community cohesion and participation reduce stigma [39–41] However, no study has directly linked community cohesion with reducing cancer stigma in policy opposition. These findings emphasize the significance of contextual factors in shaping attitudes and perceptions related to cancer stigma across different settings.

The study revealed a significant inverse association between education and cancer stigma among participants. Similar findings were reported in studies conducted in Ireland [42] and China [43] where individuals with lower levels of education had higher cancer stigma scores. These findings imply that the link between education and cancer stigma may exist across different cultural contexts and regions. Those with lower levels of education may have limited access to health literacy and information on health-related matters, potentially leading to misconceptions and misinformation.

Our findings suggest a positive association between older age and cancer stigma among women in Nepal, which is in contrast to studies conducted in England, [10] China, [44] and Kenya [20]. This difference might be explained by the comparatively lower exposure to social media platforms. Specifically, 2.2% of Nepali women aged 50 and above use social media [45]. In the UK, media exposure is considerably higher, with 50% of individuals using social media at 50, [46] while in China, 41.5% of social media users are aged above 40 years [47]. Social

media might reduce stigma, fostering awareness, empathy, and community support by disseminating accurate information, personal stories, and advocacy efforts [48]. Other mass media sources for information in Nepal include radio and television; however, there is a lack of literature regarding their use among women in Nepal. Another common source of information on cancer and other health issues is Female Community Health Volunteers through their Health Mother Group meetings. Women aged 35 to 45 are more likely to attend these meetings compared to older age groups [49] This highlights the need for targeted educational interventions to improve knowledge and awareness about cancer among older women in Nepal.

This is the first quantitative study conducted in Nepal among healthy adult women to identify the factors associated with cancer stigma. A previous qualitative study in Nepal focused on a limited pool of people with cancer, exploring only a few dimensions of stigma [50]. The current research's strength lies in applying a quantitative approach that introduces objectivity to measure diverse stigma domains among healthy women who have not experienced cancer themselves. In addition, we used CASS, a standardized tool validated in Nepal, to collect our data to estimate the factors associated with cancer stigma [27]. We utilized multivariable linear regression models, adjusting for potential confounders, including age, education, ethnicity, occupation, parity, and religion, which helps to eliminate alternative explanations of our findings.

This study has a few limitations. First, response bias is possible due to the interviewer-administered nature of the surveys. Respondents might have downplayed negative emotions in their responses, possibly causing an underreporting of their experiences and, consequently, our stigma score [51]. Second, convenience sampling techniques may introduce selection bias, limiting our findings' generalizability to the Nepalese female population. Future research with a random sample of women is needed to confirm our findings. Third, the study's cross-sectional design provides a snapshot of cancer stigma at a specific time. It cannot determine changes in stigma over time. Fourth, this study employed CASS items to evaluate cancer stigma in general. Nevertheless, stigma may vary across different cancer types [52]. Consequently, future research should strive to explore the stigma associated with specific cancer types and their determinants.

## Conclusion

In conclusion, this study found that overall cancer stigma was low, with a mean stigma score of 2.6 (SD = 0.6), indicating its presence among women in a suburban area of central Nepal. Cancer stigma was highest in the domains of personal responsibility, severity, and financial discrimination. Stigma was higher among older women and lower among those with higher education. Since stigma may affect participation in cancer screening, stigma-reduction interventions targeting older and less educated women are recommended. In conclusion, this study found that overall cancer stigma was low, with a mean stigma score of 2.6 (0.6), indicating its presence among women in a suburban area in central Nepal. Stigma may impact engagement in cancer screening efforts, so stigma reduction intervention focusing on older and less educated women is recommended.

## Supporting information

**S1 Dataset.**
(DOCX)

**S1 Data. Data collection tool.**
(DOCX)

**S1 File. "Association between cancer stigma and cervical cancer screening uptake among women of Dhulikhel and Banepa, Nepal".** As a part of the co-submission. (PDF)

## Author Contributions

**Conceptualization:** Bandana Paneru, Aerona Karmacharya, Sunila Shakya, Sangini S. Sheth, Anne Stangl, Archana Shrestha.

**Formal analysis:** Bandana Paneru.

**Investigation:** Bandana Paneru, Aerona Karmacharya, Soniya Makaju, Diksha Kafle, Priyanka Timsina.

**Methodology:** Bandana Paneru, Aerona Karmacharya, Soniya Makaju, Diksha Kafle, Lisasha Poudel, Sushmita Mali, Sunila Shakya, Donna Spiegelman, Sangini S. Sheth, Anne Stangl, Archana Shrestha.

**Project administration:** Archana Shrestha.

**Resources:** Sunila Shakya.

**Supervision:** Diksha Kafle, Sunila Shakya, Archana Shrestha.

**Validation:** Archana Shrestha.

**Writing – original draft:** Bandana Paneru, Soniya Makaju, Lisasha Poudel, Priyanka Timsina, Kalpana Chaudhary.

**Writing – review & editing:** Lisasha Poudel, Sushmita Mali, Priyanka Timsina, Namuna Shrestha, Dinesh Timalsena, Kalpana Chaudhary, Niroj Bhandari, Prasanna Rai, Sunila Shakya, Donna Spiegelman, Sangini S. Sheth, Anne Stangl, McKenna C. Eastment, Archana Shrestha.

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
