## [Decision Letter · Decision Letter 0]

21 Jun 2024

PONE-D-24-08679Socio-Economic Factors Associated with Cancer Stigma among Apparently Healthy Women in Semi-urban Nepal.PLOS ONE

Dear Dr. Paneru,

Thank you for submitting your manuscript to PLOS ONE. After careful consideration, we feel that it has merit but does not fully meet PLOS ONE’s publication criteria as it currently stands. Therefore, we invite you to submit a revised version of the manuscript that addresses the points raised during the review process.

We look forward to receiving your revised manuscript.

Kind regards,

Achyut Raj Pandey

Academic Editor

PLOS ONE

Journal Requirements:

Reviewer's Responses to Questions

**Comments to the Author**

1. Is the manuscript technically sound, and do the data support the conclusions?

Reviewer #1: No

Reviewer #2: No

2. Has the statistical analysis been performed appropriately and rigorously? 

Reviewer #1: No

Reviewer #2: Yes

3. Have the authors made all data underlying the findings in their manuscript fully available?

Reviewer #1: Yes

Reviewer #2: Yes

4. Is the manuscript presented in an intelligible fashion and written in standard English?

Reviewer #1: Yes

Reviewer #2: No

5. Review Comments to the Author

Reviewer #1: Thank you for providing opportunity to review the paper. Some comments regarding this manuscript are as follows:

1. Title says semi-urban Nepal which miss lead authors because it seems to be carried out in two purposively selected municipalities of Nepal and this result is not generalizable to all semi-urban area as you have mentioned in the limitation section of the manuscript.

Methods:

2. Please check if the regression analysis is multivariable or multivariate ? [Helping document: https://www.ncbi.nlm.nih.gov/pmc/articles/PMC3518362/]

3. Please clarify about the how variables were selected for multivariable regression analysis.

4. Have you performed the diagnostics to test for the assumptions of the linear regression analysis. If yes, please write in details about how you performed the diagnostics. In addition, How was the distribution of the residuals?

Have you checked for the multicollinearity ?

5. You have included SD in the results section. Please write its full form in data analysis section and the bottom of the respective tables in which they are used.

6. Consider writing SD in uniform way throughout the manuscript [abstract and text part used ± whereas line 170 and table 1 used mean(SD)]

7. variables like at least one CCS screening can be a confounder. Why it is not adjusted in multivariable regression

8. There are chances of clustering of data, community of people sharing common characteristics. it is better to address the clustering of the data.

Results:

9. Line 197 CI can be opened in line 165

consider writing p-value in uniform way throughout the paper [for example: there is different in line 202 and line 198 ]

10. Reference 31 seems to be the first paper linked to this paper. The sample size is 426. Table 1 from published paper(https://journals.plos.org/plosone/article?id=10.1371/journal.pone.0285771) doesn't match with the table 1 of this paper. For example: there were 184 brahmin/chhetri/sanyashi/thakuri but later it is 182. similarly there are slight differences between two papers. I think the analysis has to be redo after correcting the data .

11. Clarify what other occupations or ethnicity includes in methods or at the bottom of the tables.

12. Readers would like to read the crude CASS score for each categories of selected variables along with the overall CASS score

Discussion:

13. Line 212 second part to line 217 is confusing. I think you are summarizing the result but abruptly explanation of the domains comes.

14. Line 226 -line 230: You have been comparing CASS score between Nepal and UK and abruptly you are saying that stigma varies across diseases... This is irrelevant in this paragraph.

15. You are comparing with High income countries. It would be better if it is possible to compare with the low and middle income countries.

16. Line 234: What exact difference in health financing and community structure that resulted in low stigma score in policy opposition. Is there any specific study that reported that difference in health financing and community structure affecting stigma score

17. Line 265: multivariate or multivariable??

References

18. Reference 29 is not valid. Need to cite the original source.

19. Reference 41 is to be corrected

20. Reference 28 to be corrected

Reviewer #2: While reviewing the paper, I came across two other similar papers published from the same data set by the same authors.

(1) Shrestha A, Stangl AL, Paneru B, Poudel L, Karmacharya A, Makaju S, et al. Validation of

361 the Cancer Stigma Scale in Nepalese Women. Asian Pac J Cancer Prev. 2023 Jan

362 1;24(1):207–14.

(2) Paneru B, Karmacharya A, Bharati A, Makaju S, Adhikari B, Kafle D, et al. (2023) Association between cancer stigma and cervical cancer screening uptake among women of Dhulikhel and Banepa, Nepal. PLoS ONE 18(5): e0285771. https://doi.org/10.1371/journal.pone.0285771

The first paper presents the validity of cancer stigma scale in Nepalese context while the second paper measures the prevalence of cancer stigma and its association with cancer screening uptake. This manuscript also measures the prevalence of cancer stigma, and further identify its associated socio-demographic factors. Authors have not cited the second paper which looks very similar to this manuscript. The prevalence of cancer stigma is presented in both papers; the only difference is that the figures are reported in either percentage or mean. The socio-demographic characteristics of the participants (Table 1) are presented in all the three papers which is same and hence redundant. The authors could have mentioned that the findings are reported in previous work and hence avoid redundant information in the manuscript.

The authors must provide a sound scientific rationale for the submitted work and clearly reference and discuss the existing literature. The authors need to provide adequate justification for the novelty of their work.

Minor comments

Abstract background: Mention it as either general population or general female population in Nepal

Abstract results: could you present the measurement scale as well? You can present the scale in methods. This will help to understand the mean score in practical terms

Abstract conclusion: the sentence is not clear. Better to rephrase as 'While..., some subdomains of stigma were higher among ---in Nepal'

Introduction: the year 2020 is redundant (1st paragraph)

Introduction: who does the non-patient refer to? A more clarity is required. (3rd paragraph)

Introduction: Could you mention the full form of CASS initially? You can abbreviate afterwards (last paragraph)

Method: better to present how many items were under each of the six domains (Measures)

Results: How was age treated in the linear regression model? could you make it more clear taking readers into attention?

Discussion: what is your notion of categorizing higher and lower levels of cancer stigma when you are presenting means of the sub-domain? (1st paragraph)

Discussion: what about other mass media platforms? the sources of information regarding cancer may be varied in Nepalese context (5th paragraph)

Conclusion: The conclusion that prevalence of cancer is low needs further explanation while referring to the mean scores.

Proper use of abbreviation is recommended throughout the paper.

6. PLOS authors have the option to publish the peer review history of their article (what does this mean?). If published, this will include your full peer review and any attached files.

Reviewer #1: No

Reviewer #2: No

---

## [Author Response · Author response to Decision Letter 0]

25 Jul 2024

Jul 19, 2024

Dear Reviewers,

Thank you very much for your valuable comments. They were very helpful in improving this manuscript. I have incorporated the suggested changes into the manuscript to the best of my ability. Please find the response to the comments.

Reviewer 1

Title

1. Title says semi-urban Nepal which miss lead authors because it seems to be carried out in two purposively selected municipalities of Nepal and this result is not generalizable to all semi-urban area as you have mentioned in the limitation section of the manuscript.

Response: Thank you very much for your response. We will revise the title as follow; “Socio-Economic Factors Associated with Cancer Stigma among Apparently Healthy Women in two selected municipalities Nepal

Methods

2. Please check if the regression analysis is multivariable or multivariate ?

Response: That was an error. Thank you for noticing. We have revised the term multivariate to multivariable.

3. Please clarify how variables were selected for multivariable regression analysis.

Response: Variables were selected for multivariable regression analysis based on previous literatures and prior knowledge. We have added this information in the methods section

ADD THE INFO

4. Have you performed the diagnostics to test for the assumptions of the linear regression analysis. If yes, please write in details about how you performed the diagnostics. In addition, How was the distribution of the residuals?

Have you checked for the multicollinearity ?

Response: Yes, we performed the diagnostics for linear regression. We checked for linearity, independence, normality, and equal variances of the residuals; and collinearity We have added this information in the method section as follows. 

We checked the assumptions of linearity, independence, normality, and equal variance for linear regression in our analysis. We used a scatterplot of residuals versus predicted values to assess the linearity and equal variance assumptions. To check for normality, we examined a histogram and a normal probability plot of the residuals. We used the GEE model to address clustering (dependence) by ward, which has been addressed in the methods. Additionally, to detect multicollinearity, we used the variance inflation factors (VIF) method. The mean VIF was around 2.10, indicating that multicollinearity was not problematic.

5. You have included SD in the results section. Please write its full form in the data analysis section and the bottom of the respective tables in which they are used.

Response: Thank you very much for your comments. We have added the full form of SD as Standard Deviation in data analysis section and the bottom of the respective tables 

6. Consider writing SD in uniform way throughout the manuscript [abstract and text part used ± whereas line 170 and table 1 used mean(SD)]

Response: Thank you very much. We have made mean (SD) in a uniform way throughout the manuscript. 

7. variables like at least one CCS screening can be a confounder. Why it is not adjusted in multivariable regression

Response: We didn't adjust for screening uptake because it could serve as a mediator in the relationship between independent variables and cancer stigma. For example, for age, older individuals might have higher screening rates due to more frequent healthcare interactions, which could reduce stigma through increased awareness and education. Adjusting for this mediator could lead to over-adjustment bias.[1]

8. There are chances of clustering of data, community of people sharing common characteristics. it is better to address the clustering of the data.

Response: Yes, will utilize Generalized Estimating Equations (GEE) with multivariate linear regression, exchangeable working correlation, and robust variance to account for the clustering effect of community people on stigma scores, using the ward level as the clustering level. However, the results from the GEE analysis were similar to those obtained from the multivariable linear regression. The revised table.

Response: We have updated this section and summarize the main findings in the first paragraphs as follows:

Our study found a low level of general cancer stigma among apparently healthy women in semi-urban Nepal. However, we observed high levels of cancer stigma in the domains of personal responsibility, severity, and financial discrimination. The cancer stigma score was lowest in the policy opposition domain. Cancer stigma scores were higher among older individuals and those with lower levels of formal education.

9. Line 197 CI can be opened in line 165

consider writing p-value in uniform way throughout the paper [for example: there is different in line 202 and line 198 ]

Response: Thank you very much for your comments. We will write p-value in a uniform way throughout the paper.

10. Reference 31 seems to be the first paper linked to this paper. The sample size is 426. Table 1 from published paper(https://journals.plos.org/plosone/article?id=10.1371/journal.pone.0285771) doesn't match with the table 1 of this paper. For example: there were 184 brahmin/chhetri/sanyashi/thakuri but later it is 182. similarly there are slight differences between two papers. I think the analysis has to be redo after correcting the data .

Response: This was an error. We will rectify this. 

11. Clarify what other occupations or ethnicity includes in methods or at the bottom of the tables.

Response: We have clarified other occupations at the bottom of the tables. Other occupations include cleaner, helper, tailor, labor, parlor, and teacher.

12. Readers would like to read the crude CASS score for each category of selected variables along with the overall CASS score

Response: We have added the crude CASS score for each domain and over all CASS score with the category of the selected variables in Table 3 in revised manuscript.

13. Line 212 second part to line 217 is confusing. I think you are summarizing the result but abruptly explanation of the domains comes.

Response: We have updated this section and summarize the main findings in the first paragraphs as follows:

Our study found a low level of general cancer stigma among apparently healthy women in semi-urban Nepal. However, we observed high levels of cancer stigma in the domains of personal responsibility, severity, and financial discrimination. The cancer stigma score was lowest in the policy opposition domain. Cancer stigma scores were higher among older individuals and those with lower levels of formal education.

14. Line 226 -line 230: You have been comparing CASS score between Nepal and UK and abruptly you are saying that stigma varies across diseases... This is irrelevant in this paragraph.

15. You are comparing with High income countries. It would be better if it is possible to compare with the low and middle income countries.

Response: Thank you very much for your feedback. There are very limited studies on cancer stigma in low- and middle-income countries. We have included literature from India and Nepal in the discussion section.

 We have revised this paragraph as follows and incorporated the study from Nepal and India . Discussion is revised as follows;

In second paragraph

In previous studies exploring cancer stigma, consistently low levels of stigma have been reported, though these levels vary across different domains. Similar to our findings, a recent community-based study conducted in India[2] and a hospital-based study in Nepal [3] reported high stigma in the domains of severity, awkwardness, and personal responsibility, with the lowest levels in policy opposition. Additionally, a cross-sectional study conducted in the UK showed higher cancer stigma in the domains of personal responsibility and severity, similar to our study. [4] In contrast to women from the UK, our study population exhibited a higher total mean cancer stigma score across five sub-domains: severity, awkwardness, financial discrimination, personal responsibility, and avoidance. However, our study participants demonstrated a lower mean stigma score in the domain of policy opposition compared to English women, indicating less support for government funding toward cancer care and treatment.[4,5] 

16. Line 234: What exact difference in health financing and community structure that resulted in low stigma score in policy opposition. Is there any specific study that reported that difference in health financing and community structure affecting stigma score.

Response: We have added the explanation in the discussion section with citation 

17. Line 265: multivariate or multivariable??

Response: Has been addressed in number 2. Multivariate is revised to multivariable throughout the paper.

References

18. Reference 29 is not valid. Need to cite the original source.

Response: revised citation as follow;

Ministry of Health and Population. Female Community Health Programme [Internet]. [cited 2024 Jul 11]. Available from:https://mohp.gov.np/program/female-community-health-programme/en

19. Reference 41 is to be corrected

Response: Rectified as follow;

Acharya U. Nepal Social Media Users Survey 2021. 2022 May; Available from: https://research.butmedia.org/wp-content/uploads/2022/06/SocialMediaSurvey_Nepal_2021_CMR.pdf

20. Reference 28 to be corrected

Response: Rectified as follow;

Brooks S. Comparing Two Proportions – Sample Size [Internet]. [cited 2022 Jul 31]. Available from: https://select-statistics.co.uk/calculators/sample-size-calculator-two-proportions/

Reviewer 2

While reviewing the paper, I came across two other similar papers published from the same data set by the same authors.

(1) Shrestha A, Stangl AL, Paneru B, Poudel L, Karmacharya A, Makaju S, et al. Validation of

361 the Cancer Stigma Scale in Nepalese Women. Asian Pac J Cancer Prev. 2023 Jan

362 1;24(1):207–14.

(2) Paneru B, Karmacharya A, Bharati A, Makaju S, Adhikari B, Kafle D, et al. (2023) Association between cancer stigma and cervical cancer screening uptake among women of Dhulikhel and Banepa, Nepal. PLoS ONE 18(5): e0285771. https:/

The first paper presents the validity of cancer stigma scale in Nepalese context while the second paper measures the prevalence of cancer stigma and its association with cancer screening uptake. This manuscript also measures the prevalence of cancer stigma, and further identify its associated socio-demographic factors. Authors have not cited the second paper which looks very similar to this manuscript. The prevalence of cancer stigma is presented in both papers; the only difference is that the figures are reported in either percentage or mean. The socio-demographic characteristics of the participants (Table 1) are presented in all the three papers which is same and hence redundant. The authors could have mentioned that the findings are reported in previous work and hence avoid redundant information in the manuscript.

The authors must provide a sound scientific rationale for the submitted work and clearly reference and discuss the existing literature. The authors need to provide adequate justification for the novelty of their work.

Response: Thank you very much for your comments. Our team did two cancer stigma studies in the same study population.The first paper focused on the validity of the tool in the Nepalese setting. The second paper aims to assess the association between cancer stigma and screening uptake, adjusting for other socio-demographic variables. The current paper aims to determine the cancer stigma score for each item, each domain, and the overall score. The novelty of this study lies in its focus on quantitatively measuring cancer stigma within the non-cancer population, a topic that has been sparsely explored in previous research. Additionally, prior studies have not provided detailed assessments of stigma associated with individual items and domains. This information will be valuable in designing targeted stigma reduction interventions by identifying specific areas that need attention.

This is added in the introduction section in the revised manuscript. 

Minor comments

Abstract 

1. background: Mention it as either general population or general female population in Nepal

Response: This is the general female population. We have updated this in our background.

2. results: could you present the measurement scale as well? You can present the scale in methods. This will help to understand the mean score in practical terms

Response: Thank You very much for your comments. We have explained briefly about it in the method section. 

conclusion: the sentence is not clear. Better to rephrase as 'While..., some subdomains of stigma were higher among ---in Nepal'

Response: Conclusion is rephrased as follows;

While overall cancer stigma was low, some domains of stigma were higher among women in a suburban area in central Nepal; thus, indicating that cancer stigma persists in this region despite its low overall prevalence.

Introduction:

the year 2020 is redundant (1st paragraph)

Response: Revised 1st paragraph removing 2020 repeatedly. 

 who does the non-patient refer to? A more clarity is required. (3rd paragraph)

Response: 

Revised third paragraph as follows;Although people often stigmatized cancer, there is a limited exploration of public stigma in healthy general female populations.

Revise first paragraph as follows; Cancer is the leading cause of death worldwide, with age-standardized mortality of 100 per 100,000 population in 2020.[9] The majority of cancer deaths occur in low- and middle-income countries.[9] In Nepal, the age-standardized cancer incidence rate is 80 per 100,000 with a mortality rate of 54 per 100,000.[9] Despite recent improvements in treatment and survival, cancer is still a stigmatized disease,[10–12] and one of the most feared illnesses.[13]

Introduction: Could you mention the full form of CASS initially? You can abbreviate afterwards (last paragraph)

Response: Thank You very much for your comments. We have now provided the full form of CASS initially as Cancer Stigma Scale and abbreviated afterwards.

Method: better to present how many items were under each of the six domains (Measures)

Response: Thank You very much. We will add the number of items under six domains in the method section. Revised measures as follows;

The CASS consists of 25 items that assess six domains: (a) awkwardness-5 items, which measures how comfortable people feel around someone with cancer; (b) severity-5 items, which evaluates the expected severity of cancer consequences and the likelihood of recovery; (c) avoidance-5 items, which assesses the extent to which people avoid cancer patients and maintain physical distance from them; (d) personal responsibility-4 items, which determines how much a person's actions contribute to their cancer; (e) policy opposition-3 items, which gauges the perceived responsibility of the government and the public in the care and treatment of cancer patients; and (f) financial discrimination-3 items, which measures the anticipated deprivation of benefits to cancer patients from banking and insurance services

Results: How was age treated in the linear regression model? could you make it more clear by taking readers' attention?

Response: Age was treated as a continuous variable in the regression model. We have revised the interpretation as follows to make it more clear.

There is a positive association between age and cancer stigma score. When comparing two groups with a ten-year age difference, the older group had a higher cancer stigma score of 0.18 units compared to the younger group, after accounting for differences in education, ethnicity, occupation, number of children, and religion (95% CI: 0.04-0.20; p-value=0.04).

Discussion: what is your notion of categorizing higher and lower levels of cancer stigma when you are presenting means of the sub-domain? (1st paragraph)

Response: We have revised the term higher as high and lower as low in 1st paragraph of discussion

Discussion: what about other mass media platforms? the sources of information regarding cancer may be varied in Nepalese context (5th paragraph)

Response: We have now Added other platforms as well in the fifth paragraph as follows;

Other m

---

## [Decision Letter · Decision Letter 1]

12 Nov 2024

PONE-D-24-08679R1Socio-Economic Factors Associated with Cancer Stigma among Apparently Healthy Women in two selected municipalities of Nepal.PLOS ONE

Dear Dr. Paneru,

Thank you for submitting your manuscript to PLOS ONE. After careful consideration, we feel that it has merit but does not fully meet PLOS ONE’s publication criteria as it currently stands. Therefore, we invite you to submit a revised version of the manuscript that addresses the points raised during the review process. Most of the comments are addressed satisfactorilyHowever, In the introduction section,  justification of the need of the study should be adequately explained as commented by reviewer 2. For instance, what were the findings related to cancer stigma in your previous work and what would you add in your current research? Please add in the introduction section to justify.  In addition, on which basis the authors had chosen the factors (socio-economic)  associated with stigma among the many factors? As this study aims to find out the factors associated with stigma it is important to justify the selected variables. Line 96-100 needs to rewrite/rephrase to make it more clearer and academic Could you make it more clearer again about table 3 regarding age? What do you mean by 10 years difference? It is suggested that you mention details in the data analysis section.Conclusion should be made in a separate section and it should reflect the title and aim of your study. Consider rewriting the conclusion section.

We look forward to receiving your revised manuscript.

Kind regards,

Bimala Panthee

Academic Editor

PLOS ONE

Reviewers' comments:

Reviewer's Responses to Questions

**Comments to the Author**

1. If the authors have adequately addressed your comments raised in a previous round of review and you feel that this manuscript is now acceptable for publication, you may indicate that here to bypass the “Comments to the Author” section, enter your conflict of interest statement in the “Confidential to Editor” section, and submit your "Accept" recommendation.

Reviewer #1: All comments have been addressed

2. Is the manuscript technically sound, and do the data support the conclusions?

Reviewer #1: (No Response)

3. Has the statistical analysis been performed appropriately and rigorously? 

Reviewer #1: I Don't Know

4. Have the authors made all data underlying the findings in their manuscript fully available?

Reviewer #1: Yes

5. Is the manuscript presented in an intelligible fashion and written in standard English?

Reviewer #1: Yes

6. Review Comments to the Author

Reviewer #1: (No Response)

7. PLOS authors have the option to publish the peer review history of their article (what does this mean?). If published, this will include your full peer review and any attached files.

Reviewer #1: No

---

## [Author Response · Author response to Decision Letter 1]

13 Nov 2024

Dear Reviewers,

Thank you very much for your valuable comments. They were very helpful in improving this manuscript. I have incorporated the suggested changes into the manuscript to the best of my ability. Please find the response to the comments.

1.However, In the introduction section, justification of the need of the study should be adequately explained as commented by reviewer 2. For instance, what were the findings related to cancer stigma in your previous work and what would you add in your current research? Please add in the introduction section to justify. In addition, on which basis the authors had chosen the factors (socio-economic) associated with stigma among the many factors? As this study aims to find out the factors associated with stigma it is important to justify the selected variables. 

Response: We will add an introduction section as follows. 

 Our team previously conducted two studies on cancer stigma. The first assessed the validity of the Cancer Stigma Scale (CASS) in the Nepalese context, and the second examined the relationship between cancer stigma and cervical cancer screening uptake, finding that women with higher stigma were less likely to participate in screening. To better understand this, it is important to examine the level and distribution of stigma, identify the domains where stigma is most prevalent, and analyze which socio-economic groups are more affected. Therefore, in this study, we aim to determine item-specific, domain-specific, and overall cancer stigma scores, as well as assess the socio-economic factors associated with cancer stigma. Given the limited research on socio-economic influences on cancer stigma, we selected these factors based on a review of prior literature and our professional judgment.

2.Line 96-100 needs to rewrite/rephrase to make it clearer and more academic 

Response: We will revise it as follows. 

Our team previously conducted two studies on cancer stigma. The first assessed the validity of the Cancer Stigma Scale (CASS) in the Nepalese context, and the second examined the relationship between cancer stigma and cervical cancer screening uptake. In this study, we aim to determine item-specific, domain-specific, and overall cancer stigma scores, as well as assess the socio-economic factors associated with cancer stigma.

Note: In the revised manuscript it is incorporated with comment 1. 

3.Could you make it more clearer again about table 3 regarding age? What do you mean by 10 years difference? It is suggested that you mention details in the data analysis section.

Response: Thank you very much for your feedback. We will add it as data analysis as follows. 

For age variables, we presented coefficients in 10-year increments to enhance interpretability, as one-year changes yield small, less meaningful coefficients, while 10-year groupings capture more substantial age-related trends.

4.Conclusion should be made in a separate section and it should reflect the title and aim of your study. Consider rewriting the conclusion section.

Response: We will separate conclusion and revise it as follows.

Conclusion: In conclusion, this study found that overall cancer stigma was low, with a mean stigma score of 2.6 (SD = 0.6), indicating its presence among women in a suburban area of central Nepal. Cancer stigma was highest in the domains of personal responsibility, severity, and financial discrimination. Stigma was higher among older women and lower among those with higher education. Since stigma may affect participation in cancer screening, stigma-reduction interventions targeting older and less educated women are recommended.

---

## [Editor Report · Decision Letter 2]

20 Nov 2024

Socio-Economic Factors Associated with Cancer Stigma among Apparently Healthy Women in two selected municipalities of Nepal.

PONE-D-24-08679R2

Dear Dr. Paneru,

We’re pleased to inform you that your manuscript has been judged scientifically suitable for publication and will be formally accepted for publication once it meets all outstanding technical requirements.

Kind regards,

Bimala Panthee

Academic Editor

PLOS ONE
---

## [Editor Report · Acceptance letter]

4 Dec 2024

PONE-D-24-08679R2 

PLOS ONE

Dear Dr. Paneru, 

I'm pleased to inform you that your manuscript has been deemed suitable for publication in PLOS ONE. Congratulations! Your manuscript is now being handed over to our production team.

Kind regards, 

on behalf of

Dr. Bimala Panthee 

Academic Editor

PLOS ONE